# Endothelial Dysfunction after Hematopoietic Stem Cell Transplantation: A Review Based on Physiopathology

**DOI:** 10.3390/jcm11030623

**Published:** 2022-01-26

**Authors:** Giuseppe Milone, Claudia Bellofiore, Salvatore Leotta, Giulio Antonio Milone, Alessandra Cupri, Andrea Duminuco, Bruno Garibaldi, Giuseppe Palumbo

**Affiliations:** Divisione di Ematologia e Unita’ di Trapianto di Midollo, Azienda Ospedaliera Universitaria Policlinico-San Marco, 95124 Catania, Italy; bellofioreclaudia@gmail.com (C.B.); leotta3@yahoo.it (S.L.); giulio.milone89@gmail.com (G.A.M.); alessandracupri@hotmail.it (A.C.); andrea.duminuco@gmail.com (A.D.); brunga93@gmail.com (B.G.); ga.palumbo@gmail.com (G.P.)

**Keywords:** endothelial dysfunction, transplant-associated thrombotic microangiopathy, liver sinusoidal obstructive syndrome/veno-occlusive disease, capillary leak syndrome, idiopathic pneumonia syndrome, engraftment syndrome, acute graft-versus-host disease

## Abstract

Endothelial dysfunction (ED) is frequently encountered in transplant medicine. ED is an argument of high complexity, and its understanding requires a wide spectrum of knowledge based on many fields of basic sciences such as molecular biology, immunology, and pathology. After hematopoietic stem cell transplantation (HSCT), ED participates in the pathogenesis of various complications such as sinusoidal obstruction syndrome/veno-occlusive disease (SOS/VOD), graft-versus-host disease (GVHD), transplant-associated thrombotic microangiopathy (TA-TMA), idiopathic pneumonia syndrome (IPS), capillary leak syndrome (CLS), and engraftment syndrome (ES). In the first part of the present manuscript, we briefly review some biological aspects of factors involved in ED: adhesion molecules, cytokines, Toll-like receptors, complement, angiopoietin-1, angiopoietin-2, thrombomodulin, high-mobility group B-1 protein, nitric oxide, glycocalyx, coagulation cascade. In the second part, we review the abnormalities of these factors found in the ED complications associated with HSCT. In the third part, a review of agents used in the treatment of ED after HSCT is presented.

## 1. Introduction

The endothelium is a thin structure composed of a monolayer of flattened cells, covering the inner part of blood vessels and playing a pivotal role in vascular homeostasis [1]. The endothelium is provided with molecules and mechanisms with antithrombotic and anti-inflammatory functions and a set of molecules that retain prothrombotic properties. A balance among them maintains normal antithrombotic and anti-inflammatory status [2] (Figure 1).

The shift from the basal state to the procoagulant and inflammatory condition is referred to as endothelial activation; prolonged endothelium activation will lead to endothelium dysfunction (ED).

ED refers to the inability of endothelium cells (ECs) to determine vasodilatation of the vessel wall. ED is associated with reduced nitric oxide (NO) production, increased adhesiveness of leukocytes and platelets, increased endothelium permeability, and finally, apoptosis of EC [3]. ED is believed to play an essential role in cardiovascular diseases, renal diseases, infections, liver diseases, and multiorgan failure [4]. More widely, dysfunction of the vascular endothelium has been considered a hallmark of human diseases [1]. Notably, an endothelial activation induced by cytokines may contribute to the pathophysiology of COVID 19 disease [5,6,7].

ED is also frequently found in many complications arising in transplant medicine [8]. After allogeneic HSCT, many clinical factors play pathogenetic roles in ED. Some are common to other clinical settings such as advanced age, diabetes, hypertension [9,10]. However, others such as alloreactivity, infections, immunosuppressive agents, and, in HSCT, pretransplant conditioning are specific to the transplanted patients [11,12] (Figure 2).

The term alloreactivity is widely accepted and used. Immunological reactions after an allogeneic HSCT are heterogeneous and may determine ED by different mechanisms. Class I and II histocompatibility antigens, expressed on EC, may be the targets of immunologic attack [13]. The expression of class II antigens in EC is induced by γ interferon and down modulated by fluvastatin and everolimus [14].

After solid organ transplantation, antibody-mediated rejection is believed to represent antibody and complement-dependent injury to the microvasculature. When rejection is diagnosed after solid organ transplantation, the alloimmune reaction is readily apparent from the histopathology of the transplanted organ (leukocytes infiltrate, vascular damage, complement deposition, thrombosis) [15,16]. It results in allograft dysfunction, allograft loss, and accelerated graft vasculopathy [17]. However, in the rejection setting, the immunological mechanisms may also involve a cytotoxic T-cell response or NK response [16,18].

After allogeneic HSCT, the target of an alloimmune attack can be, at least theoretically, the entire vascular tree of the recipient. For instance, and as proof, graft-versus-host disease (GVHD) is associated with endothelium damage characterized, at the immunohistochemistry level, by perivascular infiltrate of activated lymphocyte and by an increased level of von Willebrand factor (v-WF) [19]. However, at gross histopathology, the evidence of alloimmune reaction is scarce. The reasons for that paucity have not yet been clarified [20].

Although alloimmunity can be the initiating trigger, other mechanisms besides direct cytotoxicity may occur, and innate immunity may take part in tissue damage without any histologically visible cellular effector mechanism [21]. In the context of innate immunity, the release of cytokine, along with the activation of complement and Toll-like receptors, is a potent mediator of tissue damage. In determining ED, infections or administration of pharmacologic agents may also act as important cofactors.

A three steps model has been proposed [22]. Predisposition for ED may be the first step. Conditioning and tissue damage (second steps) act on this baseline status to determine subclinical ED. Finally, as the third step, alloimmunity or infections or pharmacological agents may further increase the prothrombotic/proinflammatory status, causing the full-blown clinical picture.

## 2. Factors Important in Pathophysiology of Transplant-Associated Endothelium Dysfunction

### 2.1. TNF-α and Other Pro-Inflammatory Cytokines

EC is actively involved in innate immunity mechanisms such as cytokine sensing, leukocyte interaction and activation, and the recognition of danger-associated molecules as pathogen-associated molecular patterns (*PAMPs*) and damage-associated molecular patterns (*DAMPs*). However, in addition to being an important organ in innate immunity, the endothelium is also of vital relevance in connecting adaptive immunity with the innate branch. EC can respond to inflammatory and anti-inflammatory cytokines. Pro-inflammatory cytokines such as TNF-α, IL-1, IL-6, and IFN-γ activate the endothelium and, on the other hand, when stimulated with TNF, IL-1, or endotoxin, the endothelium itself can produce TNF-α, IL-6, IL-5, IL-8, MCP-1P. Furthermore, ECs express and secrete a wide range of chemokines. These short-range molecules attract specific leukocytes subsets, allowing their extravasation across the endothelium.

All these factors influence the prothrombotic-inflammatory state of EC through intracellular signaling or by extracellular membrane-based activities [23].

TNF-α is a cytokine produced by macrophages and lymphocytes [24,25]; TNF-α is considered the orchestrator of immunity; it plays a pivotal role in endothelial activation. The effects of TNF-α on EC are multiple, apart from inducing IL-6, IL-8, and the production of chemokines. TNF-α determines an increased expression of adhesions molecules on endothelium; other effects of TNF-α are the induction of iNO, production of ROS, NADPH oxidase increase, increased thromboplastin synthesis, reduction in thrombomodulin, increased PAI-1, induction of metalloproteinase, increased angiopoietin-2, and of an alarmin called HMGB-1 [26,27,28,29,30,31]. All these factors result in endothelium prothrombotic and inflammatory activation and increased vascular permeability (Figure 3).

TNF-α levels are elevated during various complications that follow the HSCT [32,33], and TNF-antagonists are helpful in their treatment. Upon stimulation with TNF-α, endothelial cells’ signal transduction involves NF-κβ signaling associated with p38 MAPK (Figure 4).

Cytokines-driven EC activation may determine the release of cytokines and preformed molecules stored in endothelium Weibel–Palade granules (adhesion molecules, ANG-2, IL-8, HMGB-1).

These constituents will act on EC in an autocrine way, further amplifying the inflammatory state of the EC. In such a way, TNF-α may trigger an amplifying and self-perpetuating loop (Figure 5).

### 2.2. Engagement of Adhesion Molecules and Toll-like Receptors

The interplay between endothelium, leukocytes, and platelets is a well-studied mechanism for endothelium activation and the induction of procoagulant-inflammatory states [29]. Increased endothelium expression of ICAM-1, VCAM, selectin, and PECAM allows rolling, adherence, and transmigration of leukocytes [34,35]. Endothelial chemokines participate in the adherence and transmigration of specific leukocyte populations. Platelets greatly facilitate the interaction of leukocytes with the endothelium. The adhesion of leukocytes represents a strong activation stimulus on the endothelium through the production of superoxide, which inactivates endothelium-derived nitric oxide. Neutrophil activation also leads to the release of NET that will determine complement activation and deposition of C5b-9 [36]. The engagement of ICAM-1 induces the phosphorylation of cytoskeleton-associated proteins, including FAK, paxillin, p130Cas, and cortactin. ICAM-dependent phosphorylation of these proteins remodels the endothelial cytoskeleton to influence cell spreading, migration, and activation [37].

Cross-linking of VCAM-1 results in the activation of Rac1 signaling, which induces weakening of tight junctions through Rho-dependent induction of stress fibers.

Toll-like receptors (TLRs) are membrane structures able to ligate PAMPs and DAMPs. Just as with antigen-presenting cells, a complete array of Toll-like receptors is present on EC. Ligands that engage Toll-like receptors are fungal, bacterial, and viral structures such as LPS, poly I:C, MASP, HBGB-1, RNA, and DNA. Endogenous antigens may engage the TLR-2, TLR-4, and TLR-9. After transplantation, Toll-like receptor activation may derive from bacteria translocated from the intestinal wall, from cell necrosis with a consequent increase in HMGB-1, and from tissue damage induced by the alloreaction. After HSCT, different Toll-like receptor genotypes are associated with the risk of invasive aspergillosis, hemorrhagic cystitis, increased frequency of severe GVHD, relapse incidence, and mucositis [38].

Toll-like receptors’ engagement in endothelial cells increases permeability, influences adhesion molecules, and activates proliferation, migration, sprouting, and angiogenesis [39]. In EC binding of Toll-like receptors activate in EC the signal transduction terminating in NF-κβ synthesis, having, therefore, a proinflammatory effect through secretion of various inflammatory cytokines (TNF-α, IL-1, IFN-α, IFN-beta, IL-8).

### 2.3. Endothelium Activation by Antibodies

After a solid organ transplantation, EC is recognized by HLA antibodies as well as non-HLA antibodies. Non-HLA antibodies are poorly defined in their specificity and do not activate complement but may still determine graft injury.

Apart from complement deposition, antibodies against EC in patients who received a solid organ transplantation might determine EC activation. Antibody-induced EC activation is associated with an increase in the expression of leukocyte adhesion molecules and cytokine production [40].

Anti-HLA antibodies may also transduce proinflammatory signals to EC independently from complement activation, and this is a different mechanism involved in chronic rejection and post-transplant vasculopathy [41].

Further, alloimmune-mediated injury of the endothelium has been shown to lead to aberrant expression of self-proteins and subsequent generation of autoantibodies.

On the other hand, anti-EC antibodies do not always have adverse effects.

Antibodies against EC may positively affect the graft outcome, a phenomenon known as “accommodation”. It is a resistance of the transplanted organ to antibody-induced injuries. The mechanisms underlying this process may involve an antibody-induced expression of prosurvival and cytoprotective proteins or regulation of the terminal components of complement [42].

In addition to adhesive molecules and HLA antigens, EC expresses other antigens involved in EC activation by monoclonal antibodies, such as anti-CD33 and anti-CD19. In more detail, CD33 antigen is expressed on endothelial cells of the liver. On this basis, the infusion of the conjugates of CD33 with calicheamicin can determine liver toxicity and SOS/VOD [43].

The CD19 antigen is expressed in mural pericytes in the central nervous system (CNS): This may explain the CNS toxicity that, in about 10% of cases, is associated with the infusion of genetically modified lymphocytes transduced with anti-CD19 molecules [44,45]. Under this condition, adherence of lymphocytes to endothelium is facilitated, leading to EC activation and transmigration inside the CNS, where lymphocytes release soluble cytokines such as IL6 and interferon gamma [46,47]. High levels of inflammatory cytokines can activate the CNS endothelium and pericytes. Therefore, central nervous system toxicity develops from increased vascular permeability. It can manifest in different forms, from edema, localized in the cortical area to microthrombi formation [48].

### 2.4. Cadherin

Cadherin is an adhesion molecule that connects the lateral sides of EC and is involved in endothelium permeability. Angiopoietin-2 can readily increase EC permeability through the internalization of cadherin. Matrix metalloproteinase also plays a significant role in loosening the intercellular adhesion molecules [49]. Serum obtained from patients affected by capillary leak syndrome can induce the repositioning of cadherin in a HUVEC monolayer [50].

### 2.5. High-Mobility Group B-1

The protein high-mobility group B1 (HMGB-1) is an “alarmin” produced by monocytes and necrotic cells. It binds to DNA, promotes transcription and repair. HMGB-1 protein is also a potent Toll-like receptor activator. In such a way, HMGB-1 increases adhesion molecules’ expression and some inflammatory cytokines (TNF-α, MCP1, IL8) and receptors (RAGE), (Figure 6). Cytokines derived from the HMGB1-induced EC activation determine a further increase in HMGB-1, and, in this manner, an amplification of the inflammatory stimulus may be established.

HMGB-1 induces dendritic cells to secrete IL-6, which is pivotal in increasing interleukin-17 (IL-17)-producing alloreactive T cells [51].

### 2.6. Angiopoietins

Angiopoietins are glycoproteins belonging to the family of endothelial growth factors and promote angiogenesis by binding the tyrosine kinase receptor Tie-2 expressed in vascular endothelium [52]. The most investigated family members are represented by angiopoietin-1 (Ang-1) and angiopoietin-2 (Ang-2). When Ang-1 binds to Tie-2, it induces a prompt receptor autophosphorylation [53]. The crucial role exerted by the Ang-1-Tie-2 signaling pathway in vascular development has been finely demonstrated in animal models. In Ang-1^−/−^ knockout mice (and Tie-2^−/−^ mice), an impairment of vasculogenesis with consequent embryonic lethality has been observed [54]. In Ang1^−/−^ embryos, ultrastructural analyses provided evidence of a less complex vascular structure characterized by few ECs, which were poorly associated with extracellular matrix and supporting cells [55].

Conversely, Ang-2 has been demonstrated to retain antagonistic effects on the Tie-2 receptor [56]. In this regard, Ang-2 overexpression in the transgenic murine model is responsible for embryo lethality, characterized by vascular alterations comparable to those observed in Ang-1/Tie-2 knockout mice [56].

Apart from their effect on angiogenesis, the interplay between Ang-1 and Ang-2 has a pivotal role in the endothelial equilibrium among proinflammatory and anti-inflammatory states. The Tie-2 receptor keeps EC in a resting state, and it is maintained tonically activated by Ang-1.

Ang-1 is produced by pericytes and preserves the integrity of the endothelial barrier through multiple mechanisms—namely, preventing endothelium leakage upon exposition to edema-inducing agents (platelet-activating factor, serotonin, or vascular endothelial growth factor), lowering leukocyte adherence to EC, and inhibiting TNF-promoted leukocyte extravasation [57,58,59,60]. Indeed, Ang-1 negatively regulates the expression of molecules that mediate the interaction between leukocytes and endothelium cells at a transcriptional level, ICAM-1, VCAM-1, and E-selectin [61]. Moreover, Ang-1 promotes EC cell survival by upregulating the expression of the antiapoptotic protein survivin via the PI3K/Akt pathway [62,63,64].

Ang-2 interferes with Ang-1 signaling and determines endothelial cell activation and leukocyte recruitment, with a consequent increase in permeability and prothrombotic state. This increased permeability is also due to the disruption of vascular microarchitecture. In this regard, Ang-2 weakens VE–cadherin junctions, thus generating gap formation and impairing the interaction between EC and pericytes with consequent pericyte loss [65,66]. Contrary to Ang-1, Ang-2 sensitizes EC to inflammatory mediators such as TNF-α and thrombin [67,68].

Ang-2 is produced by EC, in an autocrine manner, through the secretion of Weibel–Palade granules. When secreted, Weibel–Palade granules deliver other factors such as preformed IL-8, vWF, and adhesion molecules (Figure 7).

Endothelium damage constitutes a key element in the development of multiorgan dysfunction, and thus, given that Ang-1 and Ang-2 contribute to endothelial activation, their plasmatic values are helpful biomarkers in predicting the severity of sepsis. Increased levels of Ang-2 are associated with high mortality also in ARDS and in SARS-CoV-2 infection [69]. It has been reported that the administration of Ang-1 prevents organ failure in animal models of sepsis [70].

Ang-1 is produced by pericytes, a cell population of mesenchymal ontogeny that covers and adheres to the abluminal surface of endothelial cells (ECs) in arterioles, precapillary venules, and capillaries.

Pericytes secrete many cytokines, angiogenic molecules, and growth factors. They are provided with plasticity and can differentiate into other mesenchymal cell types, such as smooth muscle cells, fibroblasts, and osteoblasts. Pericytes may thus contribute to maintaining function or tissues regeneration. Pericytes have pleiotropic roles in EC, vessel stabilization, vascular tone regulation, and local and tissue homeostasis maintenance.

Ang-1 and Ang-2 have opposite effects on pericytes. After exposure to TNF-α, Ang-1 increased survival of pericytes, whereas Ang-2 increased apoptosis in these cells [71].

### 2.7. Coagulation Cascade

Thrombin, which acts as the promoter of fibrin formation, is a potent activator of endothelial cells. When activated, the endothelial cells produce procoagulant molecules (such as the tissue factor and the von Willebrand factor), which are responsible for the activation of the coagulation cascade and the recruitment of platelets. Moreover, the balance between the t-PA and the PAI-1 during endothelial activation is switched toward inhibition of fibrinolysis. Thus, the imbalance of t-PA and PAI-1 contribute to thrombotic predisposition by reducing fibrinolysis.

A state of hypercoagulability was demonstrated after marrow transplantation, in association with a reduction in anticoagulants (protein C, protein S, and AT III), an increase in procoagulants (factor VIII, fibrinogen, and vWF antigen), and changes in fibrinolytic parameters (t-PA and PAI-1) [72,73,74,75,76,77].

Despite the evident abnormalities found in the level of coagulation factors during transplantation, the role of coagulation is considered secondary to endothelium activation and not a primitive cause of endothelium-based complications. However, it may have a role as a cofactor. Consistent with this view, genetic mutations associated with thrombosis did not increase the rate of endothelium based-complications after transplantation [78]. Changes in levels of coagulation factors have been found helpful as biomarkers for VOD [79,80].

### 2.8. Nitric Oxide

Nitric oxide (NO) is a gaseous molecule that plays a relevant role in maintaining normal endothelial function and inhibiting inflammation. NO downregulates the expression of proinflammatory cytokines and chemoattractants by inhibiting the NF-κβ pathway [81,82,83]. NO is produced from arginine by three enzymes: neuronal (nNOS), endothelial (eNOS), and inducible nitric oxide synthase (iNOS).

nNOS and eNOS are expressed in neuronal and endothelial cells, respectively. eNOS acts by allowing the synthesis of cGMP [84] with a consequent decrease in intracellular Ca^++^, which determines smooth muscle relaxation and consequent vasodilatation [85]. Moreover, eNOS also determines an increase in cGMP levels in platelets, resulting in reduced platelet activation and reduced adhesion to the endothelium [86,87]. e-NOS can inhibit apoptosis of endothelial cells by S-nitrosylating caspase and other proteins that participate in the process of apoptosis [88]. Shear stress inside a vessel is a physiological stimulus for NO production by EC [89,90].

iNOS is mainly expressed by immune cells, and its production is stimulated by several mediators such as TNF-α, IL-6, and viral or bacterial components (lipopolysaccharide) [91]. Inducible NO (iNO) acts as a potent inhibitor of viral replication. It modulates the immune response through the rewiring of macrophages from the M1 state to the M2 state, which stops the proinflammatory insult to the tissues, limiting cytokine secretion [92,93].

Inducible NO (iNO) plays a role in microcirculatory dysfunction in sepsis. Lethal septic shock is accompanied by high levels of nitric oxide synthase (iNOS) activity [94,95] and significantly elevated concentrations of nitrite and nitrate in the plasma.

Notably, NO production can be negatively affected by certain drugs used in post-transplant settings such as cyclosporin [96,97].

Several studies have suggested NO as a possible contributor to endothelial damage in transplant-related toxicities. In murine GVHD models, suppressing NO production is related to increased weight loss, reduced overall survival, and defective hemopoietic reconstitution [98]. Similarly, in rat models of the hepatic sinusoidal obstructive syndrome, decreased NO levels are involved in the physiopathology of the disease [99]. NO downregulation has also been found in transplant-associated thrombotic microangiopathy [100].

### 2.9. Glycocalyx

Glycocalyx (GC) is a thin internal layer coating of the luminal side of the endothelium; its width is about 0.2–0.4 millimicron. Soluble proteins, glycoproteins, and proteoglycans constitute GC. The proteoglycans (syndecans, glypicans) bind to glycosaminoglycans: acid hyaluronic (IA), heparan sulfate (HS), dermatan sulfate, and chondroitin sulfate. The total volume in a human body is 1,5 L [101].

Important biochemical constituents are embedded in this matrix, such as growth factors, antithrombin III, and other molecules. Albumin will adhere to GC, contributing to the formation of the endothelium surface layer (ESL).

In physiological conditions, GC will act as an endothelium protection structure. GC reduces the ability of leukocytes and platelets to adhere to EC, thus inhibiting the activation of EC. GC can sense shear stress, and in response, it will increase NO synthesis inside EC [102]. In such a way, GC plays an essential role in regulating permeability and reducing the prothrombotic and proinflammatory status of EC.

The glycocalyx is degraded by sheddases: ADAMS, matrix metalloproteinase hyaluronidase, and heparanase-1. Heparanase-1 is activated during sepsis. Heparanase activation plays a role in sepsis-associated respiratory distress. The lysis products generated from heparan sulfate and hyaluronan, such as low molecular weight hyaluronic acid, have a proinflammatory action by activating Toll-like receptor 4. When it occurs, the lysosome activation and export of lysosome content are followed rapidly by the GC’s degradation. In an experimental heart model, the application of TNF may induce rapid GC destruction. Other agents with a destructive effect on GC are LPS, bradykinin, adenosine, and C-reactive protein. Hyperglycemia may cause GC injury, possibly by generating reactive oxygen species.

Syndecan-1 increases in plasma during GC degradation and the level of syndecan-1 predicts mortality in sepsis. GC is also degraded during hypervolemia. The atrial natriuretic peptide may determine an increase in permeability of blood vessels through a direct action on GC [103].

### 2.10. Thrombomodulin

Thrombomodulin (TM) is a glycoprotein synthesized by endothelial cells and by other cells of mesodermal derivation. It has many actions that contribute to maintaining EC in an antithrombotic, anti-inflammatory state [104]. TM increases the activation rate of protein C (PC), and activated PC (APC) inhibits coagulative cascade (precisely by the inhibition of factor V-a and factor VIII-a). TM has other actions dependent on activated protein C. It detoxifies histones, interfering with neutrophil extracellular traps (NET) [105].

Moreover, in vitro studies suggested that APC exerts its anti-inflammatory actions at a gene level by modulating the expression of several genes involved in EC apoptosis and in leukocyte adhesion. In endothelial cells, APC reduces NF-κβ translocation in the nucleus, and consequently, it inhibits the expression of downstream NF-kB-depended genes, including antiapoptotic protein (Bcl-2) and adhesion molecules (VCAM-1, ICAM-1, E-selectin) [106]. In leukocytes, APC inhibits the AP-1 transcription factor reducing their release of inflammatory cytokines [107].

In addition to such mechanisms that are PC dependent, TM has other anti-inflammatory actions that are PC independent. These are based on a specific structural domain of TM, the lectin-like domain. TM on endothelial cells can bind Lewis molecules, thereby inhibiting the Lewis-dependent adhesion of leukocytes on EC. The lectin-like domain of TM may combine with the “endogenous alarmin” HMGB-1, a potent inducer of endothelial cells inflammatory response and EC apoptosis. When blocked by the TM lectin-like domain, the HMGB-1 cannot bind its receptors on EC (TLR 4 and RAGE), which are potent proinflammatory receptors. TM contributes to the inhibition of C3b by factor I [106] (Figure 8).

TM may have a physiopathologic role in thromboembolic disorders, ischemia–reperfusion syndromes, sepsis, malignant hypertension, and ARDS [108,109,110]. A single study reported that administration of recombinant TM significantly reduced acute GVHD (a-GVHD) and ameliorated OS after allogeneic hematopoietic stem cell transplantation [111]. TM has been found clinically helpful as a therapy for DIC in children [112,113,114] and ARDS [108].

### 2.11. Complement Activation

Complement activation may occur by different mechanisms: canonical, alternate, and lectin pathways. Genetic abnormalities in the alternative pathway of complement activation are frequently found in thrombotic microangiopathy syndromes.

The alternative pathway of complement activation occurs on the vessel wall and is due to the spontaneous hydrolysis of C3 molecules in C3b. The alternative pathway of complement activation requires factor B, which binds to C3b. Factor H (FH) is a regulator of the activity of the alternative complement pathway. Factor H inhibits complement activation by acting on the generation of C3b. Five-factor H-related (FHR) proteins enhance complement activation by competing with the regulators FH.

Factor I (FI) is an inhibitor of complement pathways. FI can degrade complement C3b and C4b, in the presence of factor H, C4b-binding protein, complement receptor 1, or CD46. Membrane cofactor protein (MCP) is a membrane-bound complement regulator that acts as a cofactor for the factor I-mediated cleavage of C3b and C4b. Mutations in membrane cofactor protein (MCP; CD46) may predispose to the development of atypical hemolytic uremic syndrome (a-HUS).

In a-HUS, a congenital abnormality may be found in the following genes: complement factor H, complement factor I (CFI), membrane complement protein (MCP), complement factor B (CFB), C3, and complement factor H-related (CFHR) [115].

Autoantibodies against factor H [116] may also be the basis of an acquired form of a-HUS. These abnormalities lead to uninhibited C3 convertase C3bBb on endothelium and the formation of the lytic complex C5b-9. The deletion of complement factor H-related genes (CFHR3-CFHR1) has been associated with autoantibodies against FH [117].

Multiple abnormalities in the complement, either acquired or inherited, might be relevant for the pathogenesis of atypical hemolytic uremic syndrome [118].

The lectin-like domain of TM acts as a negative regulator of the alternative pathway (AP) of complement activation by accelerating the inactivation of C3b [119]. Mutations in TM genes may be found in some a-HUS patients [120]. The lectin pathway of complement is essential in natural immunity since it is activated from antigens from different microbes (bacterial, fungi, and viruses). Lectins are divided into two families: ficolins and collectins. They bind with naturally occurring mannose-binding serine proteases that are lectin associated (MASP). The complex can bind surface glycoconjugate in pathogens (pattern recognition molecules), determining phagocytosis and activating the lectin pathway of complement.

The lectin pathway of complement is activated in diverse types of TMAs. MASP-2 levels are highly elevated in all TMA patients. Moreover, using an MVEC experimental model [121], inhibition of MASP-2 (the effector mechanism of the lectin pathway) has a beneficial effect on the injury mediated by plasma obtained from TMA patients.

### 2.12. Neutrophil Extracellular Trap (NET)

Neutrophils can extrude chromatin outside their cytoplasm, and this structure, known as “neutrophil extracellular trap” (NET), can bind histones and other proteins. In these traps, microbes, such as bacteria, fungi, and viruses, are destroyed. NET is a defense mechanism against infections. Further, NET can amplify immune responses and activate complement and coagulation. NETs are suggested to be responsible for ED by both endothelial activation and damage. In vitro experiments have demonstrated that NETs have endothelial cytotoxic effects in a time- and dose-dependent manner [122]. Recently, the association of neutrophil extracellular traps with the development of thrombotic thrombocytopenic purpura, and the hemolytic uremic syndrome has been reported [36]. NETosis can determine complement activation and deposition of C5b-9 [123,124].

### 2.13. Endothelin-1

Endothelin-1 (ET-1) is a peptide isolated from endothelial cells, with potent vasoconstrictive action. Aside from its effect on vascular smooth muscles, it retains proinflammatory properties. It has been revealed that ET-1 promotes cytokine production and leukocyte recruitment. Indeed, its expression is upregulated under endothelial stress conditions, and it is elicited by ROS, ANg-2, and thrombin. ET-1 can trigger mast cell degranulation, leading to the release of IL-6 and TNF-α [125].

Moreover, it is responsible for the release by monocytes of the neutrophil chemoattractant IL-8 [126]. In vascular endothelial cells of the brain, elevated levels of ET-1 induce the expression of the adhesion molecules ICAM, VCAM, and E-selectin [127]. ET-1 induces signaling through epidermal growth factor receptor transactivation, oxidative stress induction, rho-kinase, and the activation (ET-receptor A) or inhibition (ET-receptor B) of the adenylate cyclase/cyclic adenosine monophosphate pathway. Endotoxin, TNF-α, CSA, and IL-1 upregulate and stimulate the release of the endothelin-1; as a result, ET-1 is released at high levels in sepsis [128].

### 2.14. Microvesicles

Microvesicles may be viewed as a portion of the cell membrane released outside the cell. They are derived from many cell types, including endothelium cells, lymphoid cells, mesenchymal cells, platelets, and antigen-presenting cells.

They function as coagulation activators, transporters between tissue, mRNA, miRNA, protein, and lipid-based signaling. They are sources of free DNA and RNA. Microvesicles are involved in coagulation, monocyte activation, immunity response, and induction of tolerance [129].

In GVHD-affected patients, increased secretion of microvesicles has been demonstrated. Microvesicles containing CD3+CD4+, CD3+CD8+, and CD3+HLA-DR+ may reflect the cell-mediated immune response and be more valuable than sIL-2R for monitoring and evaluation of a-GVHD [130].

Activation of endothelial cells is followed by the delivery of microvesicles in the systemic circulation. Since they are rich in procoagulant molecules, the secretion of microvesicles can further activate endothelium.

## 3. Clinical Pictures of Endothelial Dysfunction after Allogeneic HSCT

From the clinician’s point of view, the involvement of endothelium after allogeneic HSCT is frequent and may manifest in practice, with different clinical pictures. This issue has been the object of several reviews during the last decade [12,131,132]. A number of organ-specific diseases such as SOS/VOD, IPS, CLS, ES, and TA-TMA have their pathogenesis in EC dysfunction. All these diseases may terminate in multiorgan failures (MOFs). However, there is no agreement on which clinical picture has to be considered as derived from systemic endothelial dysfunction [132]. Primarily, venous-occlusive disease, capillary leak syndrome, and engraftment syndrome are not considered by all authors as dependent on a systemic endothelial dysfunction [132]. In contrast, some authors retain that EC has a relevant role in corticosteroid-refractory acute graft-versus-host disease.

### 3.1. SOS/VOD

SOS/VOD is characterized by increased bilirubin, body weight increase due to liquid retention, and painful liver enlargement. Conditioning intensity and conditioning type play significant roles, together with age, underlying diagnosis, and the state of liver parenchyma [133]. The main histopathological findings are round-up of EC, EC detachment, and downstream embolization of EC, together with hemorrhage in Disse space and narrowing of the centrum-lobular vein [134]. The increase in body weight, which can reach 10–20% of the basal value, and the lack of response to diuretic treatment, demonstrate that, in this disease, endothelium damage is systemic. Severe forms of SOS/VOD may progress to multiorgan failure with renal, lung, or CNS toxicity, thus confirming the systemic nature of this disease. SOS/VOD is more frequent in conditions of increased HLA distance between donor and recipient, but no concomitant and overt a-GVHD is evident in most of these cases.

In SOS/VOD, nitric oxide synthase activity is reduced in liver cells [99]. HMGB-1 has been found to be involved in models of VOD induced experimentally by monocrotaline [135]. An upsurge in the level of HMGB-1 follows the administration of Monocrotaline. Activation of endothelium cells in VOD is demonstrated by an increased level of vWF, ICAM1, VLA4, and Ang-2 [136].

### 3.2. Capillary Leak Syndrome (CLS)

An increase in body weight, blood pressure reduction, tachycardia, and sudden decrease in serum albumin is the cluster of clinical abnormalities found in capillary leak syndrome. In the idiopathic form, a monoclonal immunoglobulin is frequently present in the plasma. Secondary CLS may be associated with severe infections or with the administration of pharmacological agents, such as interleukin-2, GM-CSF, gemcitabine, and monoclonal antibodies anti-CD19 and anti-CD22. High serum levels of Ang-2 and VCAM1 have been found in patients affected by idiopathic CLS.

A significant increase in body weight (>2.5%) has been reported after allogeneic HSCT in 20–30% of all patients [137]. Severe hydric retention is associated with reduced survival and a higher risk of severe GVHD [138]. Patients at risk for CLS at the start of conditioning may be identified as having a high EASIX score [139].

### 3.3. Idiopathic Pneumonia Syndrome (IPS)

IPS criteria include evidence of widespread alveolar injury with symptoms and signs of pneumonia in the absence of active lower respiratory tract infection. Diagnosis is made after the exclusion of commonly found pulmonary infections. It requires an intensive diagnostic workup, including at least a bronchoalveolar lavage. Alloreactivity toward lung tissue after HSCT in SCID mice is accompanied by signs of activation of lung EC [140]. Further, in the development of experimental IPS, an injury to the vascular endothelium has been observed [141]. Vessels in the lung are surrounded by a dense mononuclear cell infiltrate. There is apoptosis of ECs, presence of activated cytoplasmic caspase 3, and TUNEL positivity of nuclei. Cytotoxicity via the Fas-FasL pathway contributes to the development of experimental IPS. A role for TNF alpha has been hypothesized [142]. The expression of ICAM-1, VCAM-1, and eNOS are increased in lung biopsies of patients developing IPS [143].

However, over half of the patients diagnosed with IPS have a virus detected in bronchoalveolar lavage (BAL) samples [144]. The significance of these viruses in the pathogenicity of pneumonia remains unclear, although emerging evidence suggests that at least in the case of human herpesvirus 6 (HHV-6), these viruses may lead to lung injury and raises plausible concern that IPS may have been misdiagnosed in earlier studies. Alloimmune reactions toward lung tissue and infections may interact and be cofactors.

An imbalance between Ang-1 Ang-2 may also have a role in IPS. This imbalance has been found in ARDS [145]. A four-endothelial biomarker panel, including elevated angiopoietin-2/angiopoietin-1 ratio, vascular cell-adhesion molecule, and von Willebrand factor, is useful in identifying acute respiratory distress syndrome [146].

### 3.4. Engraftment Syndrome (ES)

Diagnostic criteria for engraftment syndrome, according to T. Spitzer, include major criteria (non-infectious fever, skin rash, and non-cardiogenic pulmonary edema) and minor criteria (weight gain, hepatic/renal dysfunction, or transient encephalopathy) [147]. Diagnosis is reached with the development of two or more of the previously cited symptoms within 96 h of the start of neutrophil recovery (absolute neutrophil count > 100). According to Maiolino, diarrhea is a further criterion of this syndrome. In the time frame of peri-engraftment, it is possible also to observe lung abnormalities such as diffuse ground-glass opacities, often with septal thickening and small pleural effusions [148]. ES has been described after autologous transplantation in patients mainly affected by multiple myeloma, POEMS syndrome, amyloidosis, and autoimmune diseases [149,150]. ES is, however, possible in patients affected by other underlying diagnoses.

The distinction between ES and autologous GVHD is difficult since both syndromes may present the involvement of the skin and diarrhea. Indeed, the relationship between autologous GVHD and ES is still debated, and these two clinical pictures may represent the same disease. ES has overlapping signs also with capillary leak syndrome.

ES is frequent after syngeneic HSCT [151]. ES has also been described after allogeneic HSCT. Indeed, not-infectious fever and manifestations of a vascular leak (edema with weight gain) may occur during granulocyte recovery in both the auto- and allotransplantation settings [152]. ES may precede a-GVHD, these two diseases being temporally associated, or ES may represent an initial stage of a-GVHD [153]. Therefore, in the allogeneic setting, the distinction of ES and a-GVHD is a matter of debate.

### 3.5. Transplant-Associated Thrombotic Microangiopathy (TA-TMA)

Micro-angiopathic anemia is a predominating feature. Clinical and laboratory signs are anemia, schistocytes, hemolysis, increase in LDH and decreased haptoglobin, hypertension, fever, decreased renal function, and proteinuria [154,155,156]. It may involve the kidney, the central nervous system, and the intestinal tract, and can be associated with pulmonary hypertension and serosal surface effusions.

According to Jodelle, it may present in a severe form in up to 18% of all patients in a pediatric population. TRM in patients having signs of the disease may be as high as 48%.

TA-TMA has been considered an endothelial form of a-GVHD. In most of these cases, a previous a-GVHD episode is present, or a-GVHD at the time of TA-TMA diagnosis is still ongoing, although with apparently minimal clinical signs [157]. However, TA-TMA may also appear after autologous HSCT. The pretransplantation patient’s risk factors are a-GVHD, previous transplantation, MUD donor, and myeloablative conditioning [158,159].

In some cases, calcineurin inhibitors are significant contributors, and regression of signs has been reported after their discontinuation [155], although this remains a controversial issue [159]. In some other cases, infections may play essential roles as cofactors (aspergillus, HHV6, BK, adenovirus, CMV).

Intestinal TA-TMA is characterized by abdominal pain and may cause significant gastrointestinal bleeding. Differential diagnosis from intestinal a-GVHD may be problematic. Nishida et al. showed that intestinal thrombotic microangiopathy might mimic clinically a progressive a-GVHD [160]. Histopathology of intestinal biopsy shows microangiopathic changes in the gastrointestinal vasculature. Histologic features include endothelial cell swelling, endothelial cell separation, perivascular mucosal hemorrhage, intraluminal schistocytes, intraluminal fibrin, intraluminal microthrombi, loss of glands, and total denudation of mucosa [161,162].

Laskin et al. have reported complement activation after TA-TMA in HSC transplantation. These authors found in renal biopsy C4d in the vessel [163]. Indeed, complement activation has been demonstrated in this type of TMA with an increase in C5b-9 [164,165]. Subsequently, similar complement regulatory defects to those found in a-HUS were identified in a small series of pediatric patients affected by TA-TMA [166].

Complement activation due to genetic deletion of CFI and CFH has also been demonstrated in children suffering transplant-associated thrombotic microangiopathy [167] (Figure 9). However, further confirmations of this physiopathological view are needed, in children as well as in adults.

A two-step pathogenetic process of TA-TMA may be hypothesized. Genetic abnormalities such as deficiency of complement regulatory proteins (CFI, CFH, complement genes CFHR1-CFHR3) may be predisposing factors. Subsequently, high dose chemotherapy, a-GVHD, factor H autoantibody, or infections will lead to endothelium damage and complement activation [115,159,166,168].

A reduced level of NO has also been found in TA-TMA [169]. High levels of NET predict TA-TMA; NETosi can determine complement activation and deposition of C5b-9. In TA-TMA, the immunosuppressive agents and infections are suspected of contributing as copathogens [170,171].

The concentration of soluble products derived from complement activation (C5b-9, C3a) may be measured in plasma. It could help in predicting the diagnosis and severity of TA-TMA [164,172,173,174].

### 3.6. Endothelial Dysfunction and GVHD

Conditioning and a-GVHD are linked to cytokine secretion and endothelium dysfunction. vWF levels measured after conditioning anticipate and predict GVHD [175]. A high level of Ang-2 at the start of conditioning and later early after transplantation also predicts GVHD and TRM. Ang-2 levels since admission are higher in patients who will develop a-GVHD. Ang-2 remains high in patients affected by corticosteroid-refractory GVHD, while it is reduced in patients responding to treatment of GVHD. Endothelium damage is an important mechanism in the physiopathology of the disease, and it is FAS driven. There are some situations in which ED manifest during severe and protracted a-GVHD. We observed severe ED in the gastrointestinal tract and the CNS following an episode of a severe form of acute GVHD.

Patients with corticosteroid-refractory a-GVHD exhibited elevated serum levels of Ang-2, sTM, HGF, and IL-8 post-transplantation, compared with patients with sensitive a-GVHD and patients without a-GVHD (Dietrich et al. 2013). A high level of Ang-2 persisting after first-line therapy is a marker of corticosteroid-refractory GVHD [176], and ED may explain gastrointestinal signs and symptoms in these patients. Luft hypothesized that endothelial cell vulnerability and dysfunction, rather than refractory T-cell activity, drive the pathophysiology of corticosteroid-refractory GVHD [22,176].

A double hit has been proposed. In patients having a high Ang-2 level at pretransplantation (first hit), the occurrence of severe GVHD (second hit) will be followed by a risk of high NRM and a poor prognosis [177]. These data underline that endothelium damage may have an essential role in GVHD pathophysiology, especially in corticosteroid-refractory patients (Figure 10).

## 4. Therapeutic Interventions for EC Dysfunction

### 4.1. Defibrotide

Defibrotide is indicated in the severe form of SOS/VOD. In the INT study, the complete remission rate that can be achieved will depend on the age and the presence of MOF, and it varies between 40% and 70% [178]. Very severe forms of SOS/VOD have an unsatisfactory outcome [179].

In vitro activity on endothelial cells has been extensively studied, and it can reduce endothelial activation by lowering adhesion molecules expression and leukocyte adherence in several experimental models [180].

Defibrotide use has been explored in TM-TMA; in a limited number of patients affected with TM-TMA, the administration of a low dosage of defibrotide induced disease remission in all cases [181]. In a small study (ClinicalTrials.gov NCT03384693), administration of defibrotide as prophylaxis of TA-TMA resulted in very low NRM. The use of defibrotide as prophylaxis for acute GVHD in patients showing high risk has been proposed since a pediatric study found a reduction in transplantation toxicity and a reduction in severe GVHD [182]. In an experimental mouse model, prophylaxis with defibrotide reduces acute-GVHD and improves survival [183].

### 4.2. Anti-complement Agents

Blocking the complement system with eculizumab is currently the most effective treatment to circumvent the poor outcome in patients with severe TA-TMA [184].

In 2014, Jodelle reported that six patients were treated using eculizumab and 4/6 responded [185]. A single-center study reported eculizumab effective in 50% of adult patients, and a-GVHD was the only factor associated with inferior results [186].

At MD Anderson, in 5 years, 10 patients received eculizumab in an uncontrolled and retrospective study [187]. The anti-complement agent was associated with a change in immunosuppression. The OS in the eculizumab-treated cohort was better, compared with patients not receiving eculizumab. After transplantation, patients require modification of the dose according to CH50.

A recent meta-analysis on 116 patients suggests that eculizumab improves overall survival and response rate in patients with TA-TMA [188]. However, randomized, controlled trials and prospective studies are needed. 

Narsoplimab is a monoclonal antibody able to inhibit MASP-2. It has been found effective in thrombotic microangiopathy of COVID-19 patients [189]. Narsoplimab has been studied in the treatment of IgA nephropathy, a disease in which the lectin pathway of complement is involved; narsoplimab reduced the progression of this kidney disease [190,191]. Narsoplimab has also been studied in TA-TMA, and preliminary results have been reported (EHA 2018).

Since these anti-complement agents have high costs, there is a need for diagnostic tests that can guide after allogeneic HSCT the selection of patients to be treated.

### 4.3. Anti-CD20 (Rituximab)

This agent is currently employed in cases of TMA not responsive to plasma exchange [192].

Moreover, it is used in the setting of TMA associated with LES [193]. Some cases of patients affected by TA-TMA and improvement after treatment with anti-CD20 have been reported [194,195,196]. In recipients after transplantation, the development of antibodies against factor H [166] might be an indication of anti-CD20 treatment.

### 4.4. Withdrawal of Calcineurin Inhibitors

Cyclosporin (CSA) modifies the endothelium, and it increases the synthesis of thromboxane A2 while decreasing the production of prostacyclin (PGI-2) [197,198].

CSA inhibits NOS [199]. Disturbances in constitutive and inducible NOS in the vascular wall may predispose to vasospasm, contributing to hypertension and vascular diseases. CSA inhibits angiogenesis induced by vascular endothelial growth factor (VEGF); VEGF activates the transcription of COX2, and CSA [175] inhibits this effect of VEGF on cyclooxygenase (Cox)-2 [200]. On this basis, withdrawal of CSA or switching to tacrolimus has been advocated in patients affected by TA-TMA [201]. However, other data do not support the usefulness of this practice [159].

### 4.5. Therapeutic Plasma Exchange

Therapeutic plasma exchange (TPE) has been widely used in treating TA-TMA since the procedure is active in TTP disease (Moschowitz’s disease). However, its efficacy in the TA-TMA setting is limited, with response varying in the literature from 25% to 75% [202,203,204]. Clinically, the improvement is observed on the serum level of LDH and transfusion requirement, but survival remains very poor [204]. In TA-TMA patients, TPE is not able to prevent evolution into chronic kidney disease [205]. Moreover, the rate of complication of TPE is significant [206,207,208].

### 4.6. Thrombomodulin

TM has been found clinically helpful as a therapy for DIC in children [112,113,114] and for ARDS [108]. After allogeneic hematopoietic stem cell transplantation, a study reported that TM administration significantly reduced a-GVHD and ameliorated OS [111]. This result has been confirmed in a subsequent study [209]. TM has been found effective, after allogeneic HSCT, in cases of TA-TMA, SOS/VOD, and ES [210,211,212,213,214,215,216].

### 4.7. Statins

They are active in various diseases based on endothelial dysfunctions, such as cardiovascular disease and rheumatoid arthritis [217]. Statins have pleiotropic effects and can increase NO production in EC [218,219]. In a mice model, statins ameliorate the histopathologic signs of GVHD injury [189]. Statins increase the levels of Ang-1 [220]. In humans, prophylaxis using pravastatin after allogeneic HSCT reduces the incidence of SOS/VOD [221].

### 4.8. Angiopoietin1 Mimetics

Vasculotides is an Ang-1 mimetic provided with an anti-inflammatory effect. In animal models, it has been found to be helpful in various conditions such as hemorrhagic shock, pneumonia [222,223], strokes [224,225], and in preventing pathologic vascular leakage [226,227].

### 4.9. Alpha-1Anti-Trypsin (A1AT)

A1AT has an inhibitory effect on the expression of genes induced by TNF-alpha in endothelial cells, thereby reducing endothelial cell activation [228]. A1AT is also able to reduce the harmful effects of heme on EC [229]. Additionally, A1AT has immunoregulatory effects and decreases the production of IL-8, IL-6, TNF-a, and IL-1b. It promotes the differentiation and expansion of FoxP3+ regulatory T cells (Tregs).

A1AT has a demonstrated role in treating corticosteroid-refractory acute GVHD. Their anti-inflammatory and immunoregulatory effects merit further studies in treating complications based on ED in the transplantation setting [230].

## 5. Conclusions and Working Hypothesis

ED is recognized in many transplant-associated complications as VOD, TAM, CLS, ES, and a-GVHD. Pathogenic mechanisms are different among these complications, and we described only some of the known factors involved. In fact, the role of other factors (such as heparanase and metalloproteinase) needs to be studied [231]. Moreover, the heterogeneity of EC across different organs is a factor that limits the generalization of the present status of the knowledge.

The spectrum of complications to which ED contributes may even be broader than presently accepted. EC dysfunction can be the basis for the increased rate of cardiovascular disease found in a later phase of HSC transplantation [157].

Furthermore, ED may have a role in other transplant-associated complications, such as posterior reversible encephalopathy syndrome (PRES). PRES is a neurologic disease characterized by brain edema, which is worsened by CSA treatment. It may be considered an endothelium disease of the vascular bed of the central nervous system [22].

The endothelium is involved in the production of T-regulatory lymphocytes [232] and hence in the generation of tolerance. Endothelium dysfunction, therefore, could have a role in those clinical situations marked by a delay in the development of tolerance, such as the so-called late-onset a-GVHD.

Moreover, ED may have a role in the physiopathology of poor marrow function. This is a severe complication arising late after transplantation. The post-transplant poor graft function is associated with damage in the hematopoietic niche [233]. Since the niche has a vascular component [234], it can be hypothesized that endothelium dysfunction may be involved in poor graft function.

All complications described here based on ED may evolve or progress to multiorgan failure syndrome (MOF), despite new and specific treatment. This may be explained by an auto-perpetuating mechanism, sustaining endothelial damage independently from the initial noxae. Endothelium activation may trigger a complex cascade of parallel inflammatory mediators that lead to end-organ damage independent of the initial mechanism of ED. Although this, as a general mechanism, can occur in other clinical settings, it can be hypothesized that alloimmunity greatly facilitates this process. Therefore, the goal is to prevent ED via augmentation of endothelial repair.

## Figures and Tables

**Figure 1 jcm-11-00623-f001:**
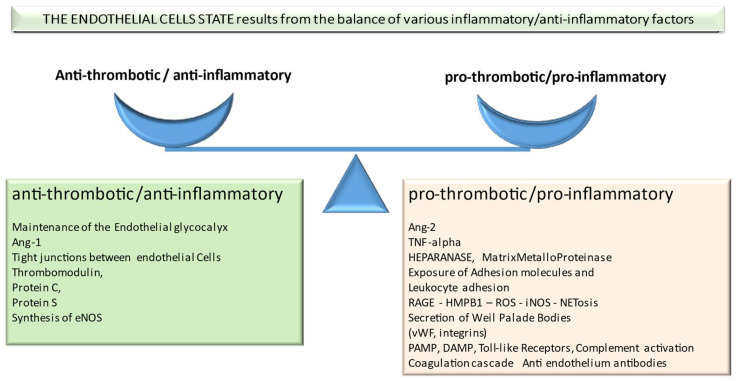
The endothelial cells state results from the balance of various inflammatory/anti-inflammatory factors.

**Figure 2 jcm-11-00623-f002:**
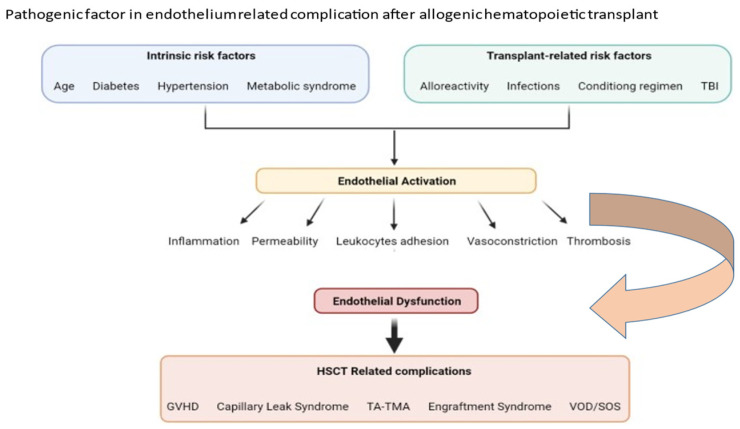
Pathogenic factor in endothelium-related complications after allogeneic hematopoietic stem cell transplantation.

**Figure 3 jcm-11-00623-f003:**
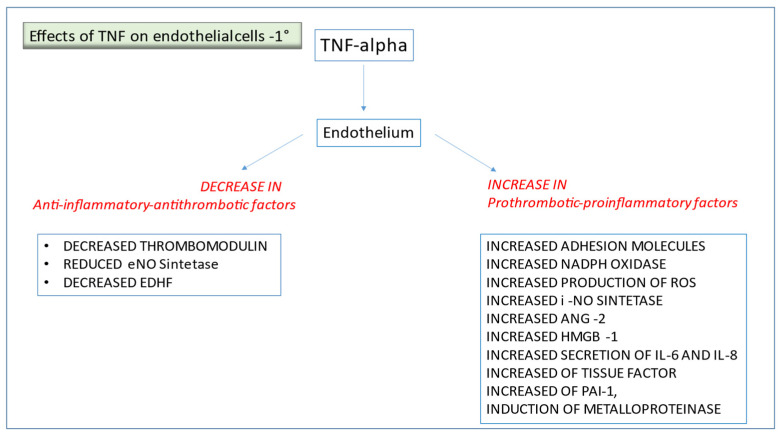
Effects of TNF on endothelial cells.

**Figure 4 jcm-11-00623-f004:**
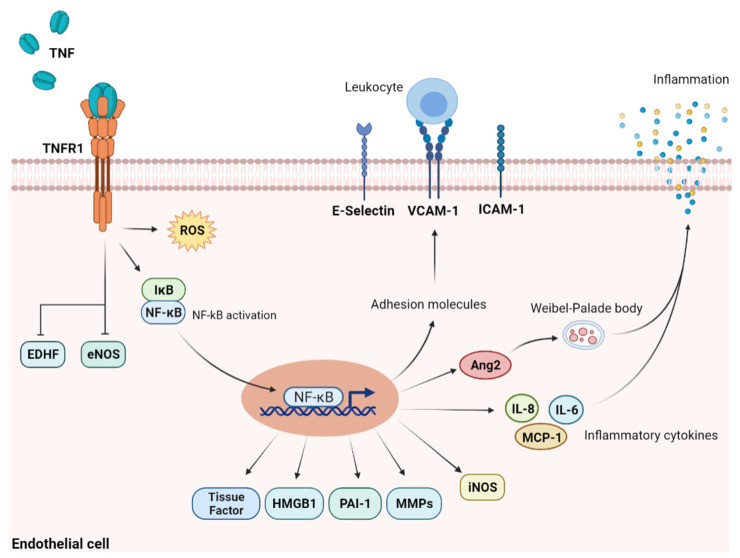
Upon stimulation with TNF-α, endothelial cells’ signal transduction involves NF-κβ signaling associated with p38 MAPK.

**Figure 5 jcm-11-00623-f005:**
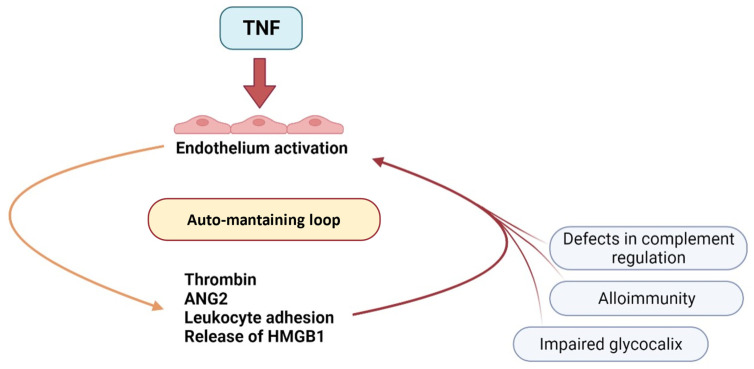
The hypothesis of an auto-maintaining loop in EC activation.

**Figure 6 jcm-11-00623-f006:**
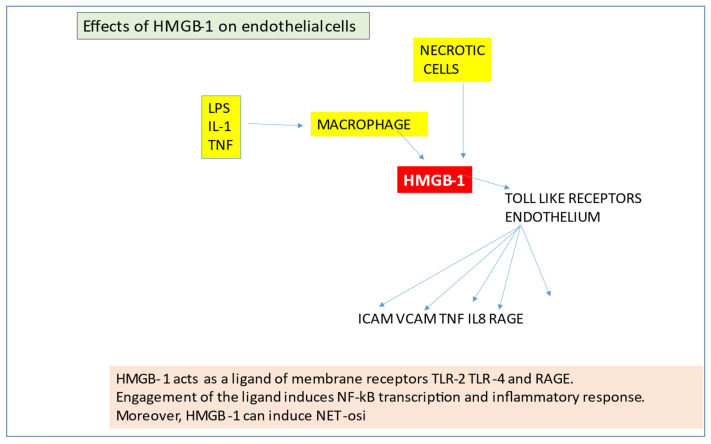
Effects of HMGB-1 on endothelial cells.

**Figure 7 jcm-11-00623-f007:**
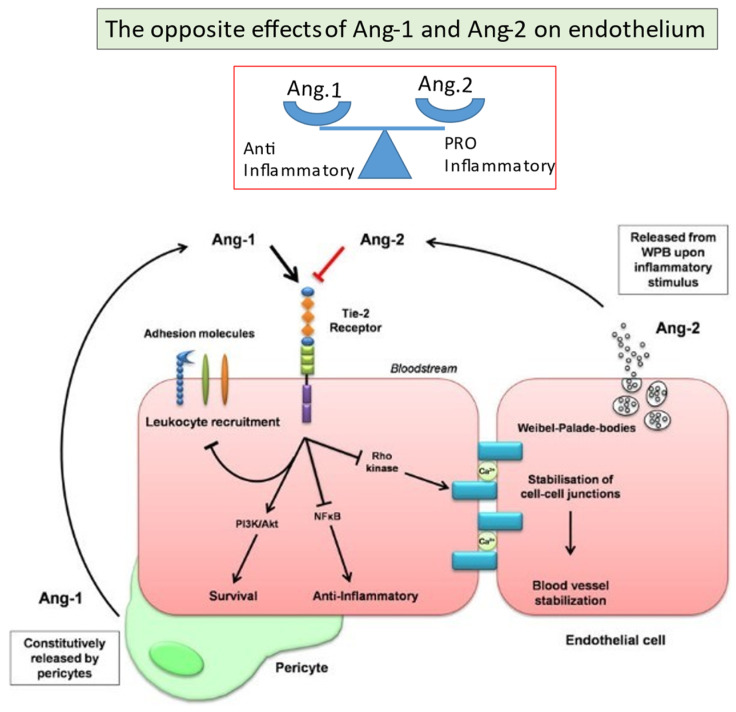
The opposite effects of Ang-1 and Ang-2 on endothelium.

**Figure 8 jcm-11-00623-f008:**
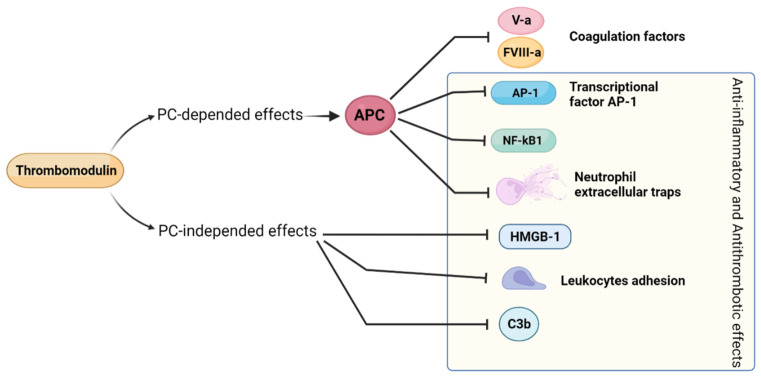
Antithrombotic and anti-inflammatory effects of thrombomodulin.

**Figure 9 jcm-11-00623-f009:**
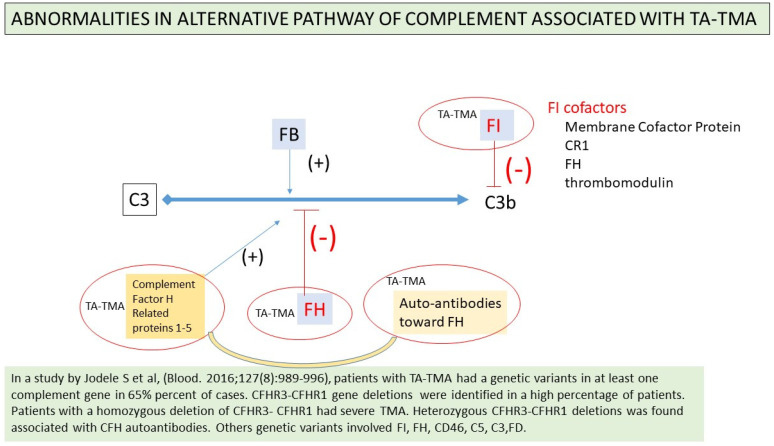
Abnormalities in alternative pathway of complement associated with TATMA.

**Figure 10 jcm-11-00623-f010:**
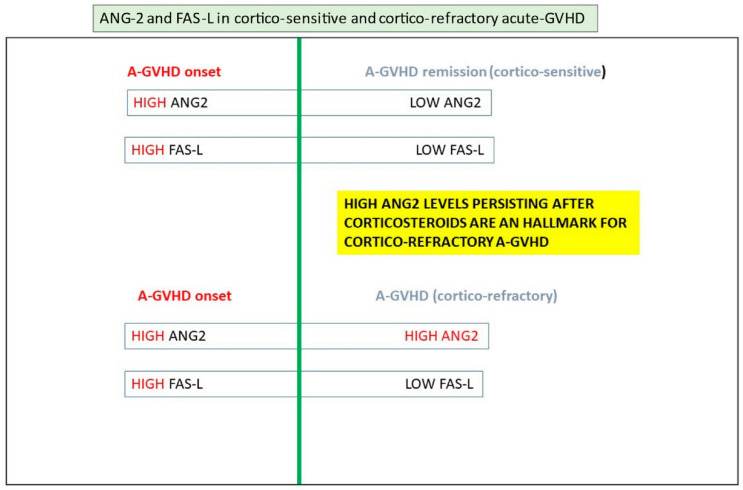
Ang-2 and FAS-L in cortico-sensitive and corticosteroid-refractory a-GVHD.

## Data Availability

Not applicable.

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
