# Peer review of "Endothelial Dysfunction after Hematopoietic Stem Cell Transplantation: A Review Based on Physiopathology"

_jcm, 2022, doi:10.3390/jcm11030623_

Round 1

Reviewer 1 Report

This is an extensive review of physiopathology endothelial dysfunction. The study analyses the main hematopoietic stem cell transplant complications and correlate them with endothelial damage risk factors.

It helps to clinicians to understand the role of endothelium in hematopoietic transplant and the possible treatment of the most important complications.

Author Response

Many thanks for your review.

Reviewer 2 Report

This review summarized the pathogenesis, clinical features and the treatment of endothelial dysfunction (ED) after hematopoietic cell transplantation. I think that this review is helpful for clinicians who take care of HCT recipients. I have some minor comments mainly associated with wordings and abbreviations as follows. I recommend to check wordings and abbreviations throughout the manuscript carefully.

1) The wordings and the usage of abbreviations such as endothelial dysfunction, hematopoietic cell transplantation and graft-versus-host disease should be uniformed throughout the manuscript.

Endothelial dysfunction: “Endothelium dysfunction” is used in Page 1 Line 35-36, 37, while “endothelial dysfunction“ in most other parts.  In addition, the usage of abbreviation is not uniformed thereafter (“ED” is used in some parts and “endothelial dysfunction” in other parts).

Hematopoietic cell transplantation: “Hematopoietic Transplantation” is used in the title and abstract. “Hematopoietic cell transplantation”  or  “Hematopoietic stem cell transplantation”  is more commonly used. It might be better to uniform following words, too; allogeneic transplantation, allogeneic transplant, allogeneic HSCT

Graft-versus-host disease: Please use uniformed abbreviations; GVHD or GvHD, a-GVHD or acute GVHD etc.

2) Figure 1 

“anti-inflammatory factors” in the box of the top of the figure seems to be printed wrong.

“Ang 2” should be corrected to “Ang-2”

3)  Page 3, Line 59-60. It seems that the new line is started wrong.

4) Page 7, Line 186 “central system toxicity” should be corrected to “central nervous system toxicity” or “CNS toxicity”.

5)  Page 15, Line 535. Authors mention that ES may represent an initial stage of hyperacute GVHD. However, hyperacute GVHD occurs at a very early stage before engraftment. In my mind, ES may represent an initial stage of “acute GVHD”.

6) Page 15, Line 553. Aspergillus should be described in Italic.

7) Page 15. It would be better to spell out GI and CMV.

8) Page 16-17, Figure 10: corticosteroid-refractory or glucocorticoid-refractory might be more common instead of “cortico-refracotry”.

9) Page 18, Line 658-662. Wording such as Cyclosporin, Cyclosporin-A and CSA should be uniformed.

10) Page 18, Line 673. It is not necessary to use capital letter in ”Chronic Kidney Disease”.

     Page 19, Line 720. It is not necessary to use capital letter in ”Posterior Reversible Encephalopathy Syndrome”.

Author Response

Many thanks for your review. All the observations have been accepted and text has been modified accordingly. Please find a word document detailing, point by point, the changes.

Reviewer 3 Report

Suggested changes to the text.

Author Response

Many thanks for your review. Number of key words has been reduced.